# Exploring the Participant-Related Determinants of Simulator Sickness in a Physical Motion Car Rollover Simulation as Measured by the Simulator Sickness Questionnaire

**DOI:** 10.3390/ijerph17197044

**Published:** 2020-09-26

**Authors:** Piotr Rzeźniczek, Agnieszka Lipiak, Bartosz Bilski, Ida Laudańska-Krzemińska, Marcin Cybulski, Ewelina Chawłowska

**Affiliations:** 1Teaching Department of Anaesthesiology and Intensive Care, Poznan University of Medical Sciences, Poland, ul. Św. Marii Magdaleny 14, 61-861 Poznan, Poland; 2Department of Preventive Medicine, Poznan University of Medical Sciences, Poland, ul. Święcickiego 6, 60-781 Poznan, Poland; alipiak@ump.edu.pl (A.L.); bilski@ump.edu.pl (B.B.); 3Department of Physical Activity and Health Promotion Science, Poznan University of Physical Education, Poland, ul. Królowej Jadwigi 27/39, 61-871 Poznan, Poland; idakrzeminska@awf.poznan.pl; 4Department of Clinical Psychology, Poznan University of Medical Sciences, Poland, ul. Bukowska 70, 60-812 Poznan, Poland; cybulski@ump.edu.pl

**Keywords:** simulator sickness, driving simulator, previous experiences, car accidents, physical motion, road safety

## Abstract

Physical motion driving simulators serve as a valuable research and training tool. Since many simulator participants suffer from simulator sickness (SS), we aimed to gain a better understanding of participant-related variables that may influence its incidence and severity. The study involved a 2-min mobile-platform car rollover simulation conducted in a group of 100 healthy adult participants. SS was measured with the Simulator Sickness Questionnaire immediately before and after the simulation. We investigated how the symptomatology of SS varies with gender, as well as with participants’ previous experiences such as extra driving training or car accidents. Although many SS symptoms occurred already before the simulation, all the symptoms except burping had a significantly greater incidence and severity after the simulation. Before the simulation, men reported disorientation symptoms more often than women, while participants with prior experiences of extra driving training or car accidents scored significantly higher in three out of four Questionnaire components: nausea symptoms, oculomotor symptoms, and the total score. The study offers interesting insights into associations between SS and prior experiences observed by means of high-fidelity real-motion simulations. More research is needed to determine the nature of these associations and their potential usefulness, for example, in helping accident survivors to cope with the distressing or even potentially disabling psychological consequences of accidents.

## 1. Introduction

Simulators of physical motion, i.e., devices or systems that imitate certain phenomena or operations of an object using its approximate model, help to observe simulation participants’ behaviours and assess their physical and psychological state. One field where simulators are extensively used is traffic and transport, where they serve as a valuable tool for research, training, and assessment purposes. Practical uses of such simulators include contributions to traffic safety research, structuring of driver training curricula, assessment of drivers’ skills, examining driver impairments, and understanding the effects of basic human limitations on driving, to name just a few [1]. Unfortunately, a large proportion of simulator participants (drivers, pilots, crew members, or passengers) suffer from simulator sickness (SS)—a syndrome experienced as a result of a simulation, and manifesting itself through symptoms such as headache, sweating, dry mouth, disorientation, vertigo, or nausea [2,3]. The syndrome seriously limits the potential usefulness of simulators as it is one of the main causes of simulation participants’ dropout. SS is similar to, but usually less severe than motion sickness (MS) [4]. It also resembles cybersickness (CS) caused by exposure to virtual reality (VR) [5]. The three syndromes—simulator sickness, motion sickness, and cybersickness—are similar with respect to the range of symptoms present, but in each syndrome, different symptoms prevail [6,7].

Numerous hypotheses have been proposed to explain why the syndromes occur. None of them has been able to fully explain the aetiology of MS/SS/CS, and their underlying cause “is likely to be a combination of these various hypotheses, together with a variable explanation” [8]. The four most prevalent theories are an evolutionary theory, a sensory conflict theory, a postural instability theory, and an eye movement theory [2,8,9,10,11]. The evolutionary theory put forward by Treisman [12] and developed by Money and Cheung [13] sees MS as a response to intoxication. The systems involved in controlling body movements are activated by slightest disruptions (neurotoxins as well as provocative motion cues). The emetic response, then, is the body’s way of “recalibrating” itself after these disruptions [8]. According to the highly popular sensory conflict theory put forward by Reason and Brand [14], SS occurs as a result of a mismatch between the stimuli expected by the sensory system on the basis of prior experiences and the stimuli actually experienced during a simulation. In short, physiological symptoms of SS constitute “the body’s response to inharmonious sensory information reaching the so-called *comparator* in the brain” [8]. The theory of postural instability (also known as the ecological theory) proposed by Riccio and Stoffregen [15] suggests that SS results from an individual’s inability to maintain postural stability in an unfamiliar environment (in this case—a simulated environment), in which he or she has not learned to maintain balance yet. According to the eye movement theory (and the related nystagmus hypothesis) offered by Ebenholtz [16,17,18], certain simulation stimuli cause specific eye movements that lead to the tension of the eye muscles, which in turn stimulates the vagus nerve and results in the unpleasant SS symptoms.

In an effort to mitigate SS and the consequent dropout of simulation participants, researchers have developed various tools measuring the sickness and thus enabling comparisons between the affected individuals. The tools include physiological, behavioural, and self-report measures [10]. The first kind of tools measures various physiological symptoms such as cardiovascular parameters, respiration, electrodermal activity, or oculographic variables [19,20,21,22,23]. Behavioural tools include various tests of postural stability, for instance, asking a participant to stand on one leg [24]. The last group of measures includes questionnaires, with the Simulator Sickness Questionnaire (SSQ) being the first choice self-report measure [4,24,25,26,27]. It is a standardised and internationally recognised instrument termed “the gold standard for measuring SS” [28], most commonly used in SS research for almost 27 years.

## 2. Background

A number of variables have been found to affect the incidence and severity of SS, MS and CS. The variables may be broadly divided into two categories: the factors related to the simulation technology used and the individual participant-related factors. The aim of this study was to gain a better understanding of the latter category, i.e., the participant-related variables that may influence the symptomatology of SS in a real-motion car rollover simulation as measured with the SSQ, with a focus on participants’ gender and prior experiences. A considerable body of research has been devoted to the effect of such physiological variables as participants’ gender [24,29,30,31,32,33,34], age [2,26,29,33,34,35,36], health status [4,37], or ethnicity [37,38], and to some psychological ones, for example pain catastrophising [39], anxiety, or neuroticism [14,38,40,41]. At the same time, relatively few studies have explored such participant-related factors as participants’ experiences. When they have, they usually focused on the effect of either repeated participation in simulations [26,42,43,44,45] or length of real-life driving experience [41,46,47]. In addition, current research on driving-related SS/MS/CS tends to use fixed-base simulations [24,32,34,46,47], which offer fewer motion cues than real-motion ones. This is for historical and economic reasons. Real motion was the obvious focus of MS research before the advent of simulators [14,48,49,50], although it continues to be studied up until now [41,51,52,53]. Simulators and the related SS susceptibility were initially studied by the army, which was able to afford large-scale research with the use of such expensive equipment [3,45]. Then simulators came into wider use in non-military transport research [54,55], but when computers and virtual reality entered our lives and became inexpensive enough to augment or replace older generation simulators, research focus shifted to CS and its covariates [37,56,57]. Computer- or VR-based simulators often have high or at least sufficient experiential fidelity (resemblance to a simulated situation). However, advanced real-motion simulators may offer better psychological fidelity by triggering the same psychological processes that underlie a simulated activity [55]. This quality would make high-fidelity real-motion simulators particularly useful for exploring if and how SS is influenced by participant-related psychological characteristics and such prior participant experiences that entail big psychological load, for instance car accidents, collisions, and special driving training. As we have access to such a simulator, we decided to examine the potential influence of these particular experiences on SS symptomatology. To the best of our knowledge, the influence of prior participation in extra driving training or in car accidents on SS symptomatology has not been investigated to date. If such influence was found, that could set the direction of future research into the nature and practical implications of this relationship. In addition, we chose to study the effect of gender in order to check how a rollover simulation compares to the results of studies that used different simulation scenarios.

## 3. Materials and Methods

The present study is part of a larger research project devoted to analysing the phenomenon of SS. The study was conducted in accordance with the Declaration of Helsinki, and the protocol had been approved by the Ethics Committee of the Poznan University of Medical Sciences on 8 March 2018 (decision number 269/18). The study was carried out at the Interactive Safety Center (Škoda AutoLab) in Poznań, Poland. It is a venue dedicated to road traffic safety and offering physical motion car rollover and crash simulations in controlled conditions. The rollover simulations at the Center are carried out using a car simulator attached to a mobile platform. The platform, on which the Škoda Superb car is mounted and is 6.5 m long and 2.20 m wide. The weight of the empty simulator is 2500 kg. Up to four participants at a time can take part in a simulation. Participants may choose to take the driver’s seat or any of the passenger’s seats. Sensors built into the simulator allow it to be run only when all the subjects are fastened with seatbelts. All loose objects are removed from the car cabin except a few dozen light plastic balls, which are placed inside on purpose. This is to show simulation participants how such objects move around the cabin during a car crash, and to raise people’s awareness of the hazard involved. During a simulation, the car is initially in a horizontal position. Then it serves as a dynamic simulator rotating around its long axis with a short stop upside down. Because of the SS that the procedure may generate, participants are asked to cross their arms in front of their chests whenever they want to discontinue the simulation.

We chose the SSQ as our main research tool. The questionnaire was created by Kennedy et al. in 1993 [4] to assess SS. The basis for developing the SSQ was the Pensacola Motion Sickness Questionnaire (MSQ), which contained 28 symptoms serving to determine the incidence and severity of motion sickness. Kennedy et al. determined that SS differed significantly from motion sickness; the former was usually much less severe and affected a smaller proportion of the exposed population. That is why the researchers eliminated from their SS analyses the MS symptoms reported too infrequently or those that showed no change in frequency or severity before and after simulator exposures. The remaining 16 symptoms that turned out to be valid in SS studies were grouped into three symptom categories, or scales: oculomotor (O), disorientation (D), and nausea (N). Kennedy et al. advised that in using the SSQ one of the two procedures should be followed: administering either a form with questions asking about the 16 symptoms only or a modified MSQ form with all the original symptoms retained but only the 16 symptoms scored. We chose the latter protocol and included only the 16 symptoms in statistical analyses (see Appendix A).

The participants in our study were recruited by convenience sampling in partnership with the Interactive Safety Center. The facility offers various commercial driving simulations and traffic safety courses. They are advertised on the Internet, attracting customers from all over Poland. The study group comprised 100 participants randomly selected from the people who volunteered to take part in the Center’s rollover simulations on 3 consecutive days. They were all adults with a driving license, 61 men and 39 women, aged 18 to 74 years. All the participants were asked to take part in the same simulation. Our study protocol did not include a control group. The use of control groups in SSQ research is relatively rare and motivated by the need to compare SS symptomatology between the exposed and the non-exposed simulation participants [58,59]. This was not the rationale behind our study; we wanted to analyse internal group differences between the exposed participants to find out which participants’ characteristics were associated with differences in SS.

As the first step of the study (see Figure 1), all the participants were informed about the procedure and gave their informed consent for inclusion. Next, as advised by Kennedy et al. [4] and the authors of the Polish version of the SSQ [25], the participants were asked to give a short self-report assessment of illnesses and current health status. For instance, there were questions about the time since the last flight, sea voyage and activities performed in virtual reality, about the current self-perceived physical fitness, diseases of the participant, the amount of alcohol consumed within the last 24 h, medicines taken, and sleep (in terms of its quality and quantity). This part of the study served mainly to exclude all unhealthy individuals and those in other than their usual state of fitness. It was done in order to minimise both the health risk for the participants and the chances of scoring data not related to the exposure. The following volunteers were excluded: persons of poor self-reported health status; persons reporting chronic diseases, cardiac problems, or hypertension; pregnant women; persons wearing prostheses; persons with past injuries of the head, neck, cervical vertebrae, chest, abdomen, spine, intervertebral discs, or legs; and persons who reported suffering from motion sickness in the past. In addition, the self-report part included questions about the participants’ history of participation in accidents or collisions or in extra driving training.

The core of the study involved carrying out the rollover simulation. To ensure each participant’s safety, Center staff checked if all loose objects except plastic balls were out of the car and if the seatbelts were properly fastened. The simulation procedure itself lasted up to 2 min. First, the car was rotated clockwise around its long axis for 20 s. Next, the rotation was stopped with the car upside down for 10 s, during which the participant was suspended by the seatbelts. Then the car was rotated counterclockwise for the next 20 s, after which it was stopped in the initial horizontal position. Finally, Center staff opened the car, checked the participant’s safety, briefly asked about his/her health status, unfastened the seatbelts and helped him/her to get out of the car. The whole simulation, up to that point, took up to 2 min.

The main research tool, i.e., the SSQ, was administered twice: to assess the symptoms immediately (i.e., 2–5 min) before the simulation (pre-test) and again within 5 min after the end of the simulation (post-test). Each participant was asked to assess the severity of symptoms using a 4-point scale, from “none” (no symptoms), through “slight” and “moderate”, to “severe” symptoms of SS. Values from 0 (none) to 3 (severe) were then scored. The 16 symptoms assessing SS were grouped into the 3 scales. Next, symptom scores within each scale were multiplied by appropriate weights and added to obtain scale scores. In addition to the scale scores, the total score for all the 16 symptoms was also computed in accordance with the procedure described by Kennedy et al. (table 4, [4]).

To evaluate the differences between the pre-test and post-test values, the paired sample t-test or Wilcoxon signed-rank test with correction for ties and zeros was used (depending on measurement). To evaluate the differences between the two compared groups, the t-test for independent samples was used. T-tests are fairly robust against violations of the normality condition. Therefore, homogeneity of variances was taken into account in this case. ANOVA with repeated measures (terms x groups) was employed. Partial eta-squared (ηp^2^), Cohen’s *d,* and the rank-biserial correlation coefficient (*r*) were calculated to determine the effect size for particular effects. The calculations were performed using STATISTICA 13 (StatSoft, Inc., Tulsa, Oklahoma).

## 4. Results

The descriptive characteristics of the study group are presented in Table 1.

Table 2 presents statistics for particular SSQ scales and the total score reflecting the severity of SS symptoms, as well as pre-test and post-test differences in the symptom groups. The scores on the disorientation scale as well as total scores showed statistically significant differences between the pre-test and post-test parts. The scores were significantly higher in the post-test part, which means a markedly more frequent occurrence of symptoms after the simulation.

Table 3 shows descriptive statistics of individual symptoms and their pre-test vs. post-test scores. The analysis of particular symptoms showed considerable differences between most pre-test and post-test scores. In a vast majority of symptoms (except burping), the difference meant a more frequent incidence and severity after the simulation.

The next step of the analysis involved checking whether particular scale scores and the total score differed between genders. Statistically significant differences were found only in the pre-test part on the disorientation scale (*t*(98) = 2.207, *p* = 0.030, *d* = 0.468), with men experiencing the D symptoms more often than women (not shown).

In further analyses, differences in subscale scores and total scores were explored with reference to participants’ prior participation in simulations (S; *n* = 11), in various forms of extra driving training (DT; *n* = 64), and in car accidents or collisions (AC; *n* = 38). The differences are presented in Table 4.

The results may indicate that prior simulation experiences do not differentiate the participants’ reactions. However, a relatively small number of participants with such experiences (*n* = 11) does not allow for definite conclusions. Prior participation in various extra driving training such as courses or car rallies did change the participants’ reactions both before and after the simulations (pre-test: O score and total score, and a marginal effect in N score; post-test: a marginal effect in O score and total score). The participants who had undergone extra driving training scored significantly higher on these scales. Participation in accidents or collisions also produced statistically significant differences in symptom scores in both pre-test (total score and marginal effects on N and O scales) and post-test (D). Just as with the previous variable, the participants with prior accident or collision experiences had significantly higher SSQ scores.

By means of ANOVA with repeated measures, we also checked whether the particular qualitative factors (prior experiences of simulations, driving training, and accident or collisions) differentiated the participants’ reactions to the simulation as measured with the SSQ. The interaction effect checked for each SSQ scale and the factors mentioned above was significant for accident experiences (AC) only with the O scale (F(1, 98) = 6.138, *p* = 0.015, ηp^2^ = 0.06) and D scale (F(1, 98) = 4.086, *p* = 0.046, ηp^2^ = 0.04), but in a different way. On the O scale, the participants with prior accident experiences had more severe symptoms before the simulation and less severe symptoms after the simulation, compared to the participants without such experiences. On the D scale, the participants with prior accident or collision experiences reported symptoms of similar severity before the simulation, but of significantly higher severity after the simulation, compared to the participants without such experiences (see Table 4).

## 5. Discussion

In our study, the incidence and severity of almost all SS symptoms increased significantly after the simulation. All the participants reported to have at least some of the SS symptoms in the post-test. In other studies, the reported incidence of SS varied widely depending on the simulation technology and SS measurement tools. For example, SS incidence among driving simulation participants is estimated to range between 35% and 75% [60].

Our pre-test results indicate that the participants experienced SS symptoms already before the simulation, with the most prominent symptoms being fatigue, general discomfort, sweating and difficulty concentrating. Interestingly, burping was more severe before than after the simulation. Applying the SSQ both before and after the exposure is consistent with the original study design used by Kennedy [4], and pre-tests have been used as a baseline in numerous studies on MS, SS, and CS [31,36,56,61], but the psychological mechanisms behind the incidence of pre-exposure symptoms are complex and unclear. It seems that the stress associated with the expected participation in a simulation may have triggered SS symptoms and thus influenced the pre-test results [62]. In addition, the reported severity of the symptoms might have been affected by a response bias. Young et al. [63] hypothesized that administering the SSQ prior to the exposure may cause a response bias in the form of demand characteristics, i.e., a participant trying to meet an experimenter’s perceived expectation to obtain big differences between pre- and post-test results. On the other hand, some researchers reported a mitigating effect of expectation and anticipation on sickness symptoms [64,65]. Therefore, it would be useful in future research to supplement the subjective measure (SSQ) with additional more objective tools such as non-invasive measurements of vital parameters (heart rate, electrodermal activity, oculographic variables, etc.).

In the present study we used only one kind of simulator and simulation protocol. Therefore, as regards the variables influencing the incidence and severity of SS, we did not analyse those related to the simulation technology used, although numerous sources stress that various simulation-related variables such as the type, setup, and parameters of the simulator used as well as the length and scenario of a task performed during a simulation have a significant impact on the severity of particular symptoms [9,10,66,67,68,69].

Instead, we tried to explore participant characteristics that may influence individual susceptibility to SS. A number of such variables were reported to affect the susceptibility to MS/SS/CS. Participants’ gender is the most frequently studied participant-related variable. In our study, we only managed to find that gender influenced the incidence of disorientation symptoms before the simulation; men experienced them more often than women. In most of the research on gender differences, women were found to be more susceptible to sickness [24,29,31,33], but opposite findings were also reported [30].

Other studies suggested the influence of age, with older adults being generally more susceptible to SS [2,26,29,33,36]. Some researchers found a link between SS and specific movement patterns or postural instability, the latter of which was reported to be more prevalent in older adults and females [24,33,40,47,70,71]. Additional participant-related factors reportedly increasing susceptibility include lower concentration level, variable mood and energetic arousal, weaker feeling of presence in simulated environment, tendency for pain catastrophising, higher anxiety levels, neuroticism, perceptual style, Chinese ethnicity, health status, and even better aerobic fitness [15,37,38,39,40,41,61,72].

The variable which we looked at was the participants’ past experiences. Some aspects of the influence of this factor on the symptomatology of SS had also been explored by other researchers. For example, repeated participation in simulations was found to usually result in a decreased incidence of SS due to adaptation to simulation conditions [42,43]. Adaptation tended to be specific only to a particular set of conditions, and habituation to real-life experiences sometimes failed to translate into habituation to simulated conditions. This might explain why experienced pilots were found to be more prone to SS than student pilots [44,45]. In contrast, some studies comparing experienced and inexperienced drivers failed to observe more severe symptoms in the former group [46] but did note its higher simulation dropout rates [47]. Generally, driving simulations might be expected to help participants adapt to unpleasant conditions and make them less prone to SS. Due to the insufficient number of participants with prior simulation experiences in the study sample, we did not manage to test this hypothesis.

However, more than a third of our study participants had previously taken part in accidents or collisions, and nearly two thirds—in extra driving training. Both these groups noted statistically higher SSQ scores. There may be a number of possible explanations. One of them is that the more pronounced SS symptoms in these participants may point to the presence of other underlying individual characteristics that make such participants more susceptible to SS and, at the same time, more prone to accidents and collisions or more often inclined to take part in extra driving training. In such case, one consideration for future research could be the search for these characteristics, for example by means of psychometric testing, with an aim to both predict accidents and, if possible, to prevent them with the help of simulations. It might also help to discover which individuals would benefit the most from extra driving training.

Another possible explanation is that such participants previously experienced distressing situations similar to the simulated one. Consequently, they may have expected a similar course of events during the simulation to come, and this resulted in heightened anxiety and anticipatory stress reactions [73]. If this is the case, some participants may have chosen to take part in a rollover simulation in order to reconstruct the distressing event and cope with it, possibly to strengthen or regain the sense of control. However, no definite conclusions about that can be reached without asking participants about the details of their previous experiences, their motivations for taking part in driving simulations, and the incidence and severity of driving-related fear in the study group. According to a review by Taylor et al. [74], the incidence of driving-related fear in accident survivors ranges from 2 to as many as 100% of study samples, depending on the term used (driving reluctance, fear, anxiety, phobia) and the defining criteria applied. The consequences of excessive driving-related fear include adverse physical and psychological symptoms (tension headaches, myalgia, gastrointestinal discomfort, panic attacks), as well as maladaptive driving behaviours (excessive caution, avoidance, poorer control and performance while driving) [75,76]. Diverse treatment approaches to driver-related fear are available, from pharmacological, through psychotherapeutic, to technology-assisted ones [76]. A number of studies indicate that simulations can be a valuable treatment method. According to Park et al. [77], exposure therapies coupled with VR simulations are beneficial in rehabilitating patients with various psychiatric and psychological problems such as phobias, anxiety, posttraumatic stress disorder, or depression, even despite the fact that VR environment may trigger SS in therapy participants. Wald [78] investigated the efficacy of virtual reality exposure treatment (VRET) for driving phobia in 5 patients. Although she found the treatment outcome to be rather modest, she concluded that VRET would be useful as a supplement or preparatory intervention for in vivo (real-life) exposure. Similar conclusions on the usefulness of VRET, at least as a supplementary measure, were reached by other authors, also in the area of treating fear of driving and trauma after traffic accidents [79,80,81,82]. In some phobias, the effectiveness of in vivo exposure therapy seems similar to the effectiveness of VRET [83,84]. Therefore, it would be useful to test in the future whether physical motion simulations such as the one applied in our study—closer to an in vivo exposure than to VR—is as useful in reducing driving anxiety.

## 6. Limitations

The present study has a few limitations. Firstly, the use of pre-test might have caused a response bias [63]. Secondly, the study protocol involved administering only the subjective measure (SSQ). Thirdly, the variable of participants’ prior experiences was not explored in detail and calls for extended research.

## 7. Conclusions

The participants of our real-motion driving simulation reported some SS symptoms already before the simulation, which may have been caused by stress. The severity of the symptoms significantly increased after the simulation. As regards associations between participant characteristics and individual susceptibility to SS, we found gender to be significant: men experienced disorientation symptoms before the simulation more often than women. The participants’ past experiences, more specifically previous accidents or collisions as well as extra driving training, also affected SS symptoms: these participants noted statistically higher SSQ scores in three out of four SSQ components before a simulation. More extensive research is needed to explore the nature of this relationship and the potential usefulness of such findings. Several studies had explored the potential for using VR simulations in exposure therapies reducing the problem of driving-related fear. The present results are too limited to conclude that real-motion simulations could be as helpful, but we hope they will spur in-depth research in this area. This would require investigating: (1) the details of past experiences and their psychological load as perceived by simulation participants, (2) associations between these experiences and SS symptomatology, and (3) potential habituation to repeated simulations and its impact on the perceived psychological load of the past experiences. If confirmed, the association between the history of traffic accidents and more pronounced SS could perhaps pave the way for using real-motion simulations as a form of exposure therapy for those struggling with driving-related fear. For now, real-motion rollover simulators may serve as a realistic tool for sensitising drivers and passengers to such simple safety rules as fastening their seatbelts and removing all loose items from the cabin.

## Figures and Tables

**Figure 1 ijerph-17-07044-f001:**
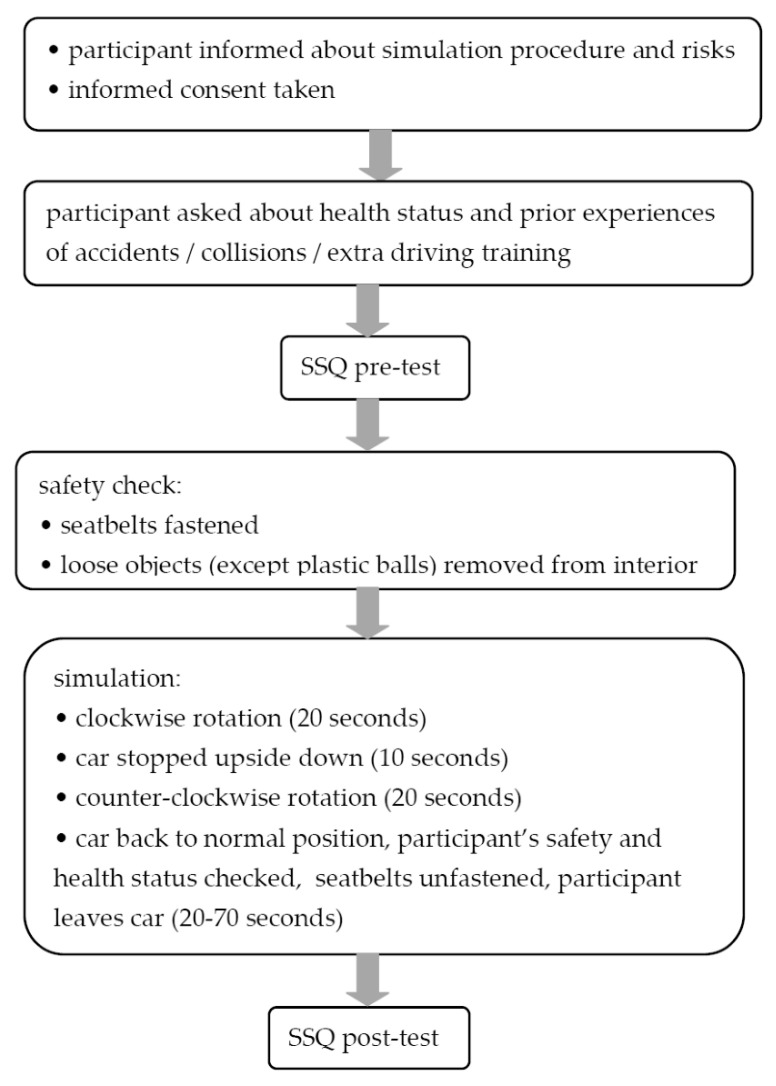
Study framework.

**Table 1 ijerph-17-07044-t001:** Descriptive statistics of the study group (N = 100).

Characteristics		M	SD
Age		39.7	13.5
Driving experience (in years)		15.4	12.3
		*n*	(%)
Gender	F	39	39
	M	61	61
Category of driving licence	B	94	94
	C	4	4
	D	2	2
Prior experiences	S	11	11
	DT	64	64
	AC	38	38

M—mean, SD—standard deviation, F—female, M—male. Driving licence categories: B—passenger cars (vehicles up to 3500 kg with up to 9 seats, with a trailer up to 750 kg), C—large vehicles (vehicles over 7500 kg, with a trailer up to 750 kg), D—buses (buses with more than 8 passenger seats, with a trailer up to 750 kg), S—simulations, DT—extra driving training, AC—accidents or collisions.

**Table 2 ijerph-17-07044-t002:** Descriptive statistics of each scale of the Simulator Sickness Questionnaire (SSQ) for pre- and post-test (the paired sample *t*-test).

SSQ Scale	Pre-Test				Post-Test				*t*(99)	*p*	Cohen’s *d*
	M	SD	Min	Max	M	SD	Min	Max			
N	28.43	18.64	0.00	85.86	32.05	24.54	0.00	143.10	−1.24	0.218	0.124
O	23.42	14.74	0.00	60.64	26.61	16.46	0.00	75.80	−1.49	0.138	0.150
D	5.85	9.73	0.00	41.76	111.92	18.23	69.60	180.96	−54.78	0.000	5.486
Total	24.27	13.53	0.00	71.06	55.76	18.79	26.18	142.12	−14.85	0.000	1.485

M—mean, SD—standard deviation, Min—minimum, Max—maximum, N—nausea, O—oculomotor, D—disorientation, Cohen’s d—effect size for repeated measures.

**Table 3 ijerph-17-07044-t003:** Descriptive statistics of particular symptoms of the Simulator Sickness Questionnaire (SSQ) for pre- and post-test (Wilcoxon signed-rank test).

SSQ Scale	Pre-Test				Post-Test				*Z*	*p*	*r*
	Me	M	Min	Max	Me	M	Min	Max			
1 General discomfort	0	0.85	0	3	1	1.28	0	3	2.736	0.006	−0.198
2 Fatigue	1	0.91	0	3	1	0.94	0	3	0.143	0.886	−0.010
3 Headache	0	0.01	0	1	0	0.08	0	1	2.073	0.038	−0.165
4 Eyestrain	0	0.39	0	2	0	0.33	0	2	0.448	0.654	0.033
5 Difficulty focusing	0	0.25	0	2	0	0.29	0	3	0.535	0.593	−0.039
6 Increased salivation	0	0.32	0	2	0	0.23	0	2	1.155	0.248	0.085
7 Sweating	0	0.69	0	3	0	0.78	0	3	0.563	0.573	−0.040
8 Nausea	0	0.03	0	3	0	0.21	0	3	2.521	0.012	−0.181
9 Difficulty concentrating	0	0.68	0	3	0	0.57	0	3	0.721	0.471	0.053
10 Fullness of head	0	0.00	0	0	0	0.00	0	0	-	-	-
11 Blurred vision	0	0.00	0	0	0	0.02	0	1	1.342	0.180	−0.100
12 Dizzy (eyes open)	0	0.00	0	0	3	2.75	1	3	8.682	0.000	−0.654
13 Dizzy (eyes closed)	0	0.10	0	1	3	2.94	2	3	8.682	0.000	−0.665
14 Vertigo	0	0.04	0	1	2	1.83	0	3	8.638	0.000	−0.641
15 Stomach awareness	0	0.02	0	1	0	0.21	0	3	2.578	0.010	−0.185
16 Burping	0	0.39	0	2	0	0.08	0	2	3.525	0.0004	0.257

M—mean, Me—median, Min—minimum, Max—maximum, Z—test statistics for Wilcoxon signed-rank test, r—rank-biserial correlation coefficient.

**Table 4 ijerph-17-07044-t004:** Differences between pre-test and post-test SSQ scores in participants with and without prior simulation experiences, extra driving training, and accidents or collisions—the results of *t*-test.

**SSQ Scale**	**Differentiating Variables**									
	**S** ***Yes***	**S** ***No***	***t*** **(98)**	***p***	**Cohen’s** ***d***	**S** ***Yes***	**S** ***No***	***t*** **(98)**	***p***	**Cohen’s** ***d***
	***M***	***M***				***M***	***M***			
**Pre-test**						**Post-test**				
N	24.28	28.94	0.78	0.437	0.194	32.09	32.05	−0.01	0.996	0.000
O	19.29	23.93	0.98	0.327	0.286	25.50	26.74	0.24	0.814	0.061
D	3.80	6.10	0.74	0.462	0.364	106.30	112.61	1.08	0.281	0.402
Total	20.06	24.79	1.10	0.266	0.285	53.72	56.02	0.38	0.591	0.175
	**DT yes**	**DT no**	***t*** **(98)**	***p***	**Cohen’s** ***d***	**DT yes**	**DT no**	***t*** **(98)**	***p***	**Cohen’s** ***d***
	***M***	***M***				***M***	***M***			
N	31.01	23.85	1.87	0.065	0.467	33.99	28.62	1.05	0.296	0.204
O	25.82	19.16	2.21	0.029	0.442	28.90	22.53	1.88	0.063	0.396
D	6.96	3.87	1.54	0.128	0.331	114.19	107.88	1.68	0.097	0.411
Total	26.76	19.84	2.52	0.013	0.575	58.26	51.32	1.79	0.076	0.399
	**AC yes**	**AC no**	***t*** **(98)**	***p***	**Cohen’s** ***d***	**AC yes**	**AC no**	***t*** **(98)**	***p***	**Cohen’s** ***d***
	***M***	***M***				***M***	***M***			
**N**	**32.89**	**25.70**	**1.90**	**0.061**	**0.377**	**33.14**	**31.39**	**0.34**	**0.731**	**0.079**
**O**	**26.93**	**21.28**	**1.89**	**0.061**	**0.343**	**23.54**	**28.49**	**−1.47**	**0.145**	**−0.312**
**D**	**6.23**	**5.61**	**0.31**	**0.761**	**0.105**	**117.22**	**108.67**	**2.33**	**0.022**	**0.497**
**Total**	**27.85**	**22.08**	**2.11**	**0.038**	**0.383**	**56.10**	**55.56**	**0.14**	**0.889**	**0.052**

M—mean, S—simulations, DT—extra driving training, AC—accidents or collisions, Cohen’s d—effect size for independent samples t-test.

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
