# Peer review of "Exploring the Participant-Related Determinants of Simulator Sickness in a Physical Motion Car Rollover Simulation as Measured by the Simulator Sickness Questionnaire"

_ijerph, 2020, doi:10.3390/ijerph17197044_

Round 1

Reviewer 1 Report

This manuscript is about exploring the participant-related determinants of simulator sickness in a physical motion car rollover simulation as measured by the simulator sickness questionnaire

The paper is of average quality. Some suggestions, the authors should consider improving the quality of manuscript:

Abstract:

Add something about the benefits of the research. [Please add-last line]

Background

Please add a heading of background and add a table with existing research, especially about the variables already used and research gap… about new variables. With at least 20-30 latest research papers. A weak literature review- specially discussion about variables and questions is missing. Discussion about already used variables and newly added variables should be there. Establish a strong research question because simulator sickness is a well-established phenomenon (See suggestions below).

[Please add this section and Table]

Materials and Methods:

A comprehensive research methodology has been designed but framework to follow the research is missing. Add framework-flowchart and write this section in stepwise pattern..Step.1, Step.2.

Interview Design:....

How questions were selected? Backup of selecting certain question (L. Review?)

What scale was adopted? Backup of selecting certain question(L. Review?)

What Analysis technique was adopted? Reason (L. Review?)

Questionnaire Design:....

How questions were selected? Backup of selecting certain question (L. Review?)

What scale was adopted? Backup of selecting certain question (L. Review?)

What Analysis technique was adopted? Reason (L. Review?)

Pilot Study of selecting certain question/questionnaire improvement?

The participants were asked to complete a simulation lasting 2 minutes. (why only 2 min time was selected???)

The study group – 100 participants randomly selected from the people who volunteered to take part in the Center’s rollover simulations – were all adults with a driving license, 61 men and 39 women, aged 18 to 74 years, of good self-reported health. (How much time it took to arrange that number of participants-Months??? Because it is not easy to have that many number for this purpose.)

Add Simulator setup images.

Suggestions to improve:

Please Add Tables along with citation in the manuscript

(a) Previously used Variables (b) New Variables to be studied (c) Previously asked Questions (d) Previously used techniques for similar type of research.

-Select one of existing techniques.

-A strong literature review will strengthen this manuscript.

-Add Simulator images to show working.

Results:

Results should be linked with research gap…. shown in the form of some statistics. [Please add]

Conclusion:

This section is missing. This section must contain implications for research, practice and/or society: Does the paper identify clearly any implications for research, practice and/or society? Does the paper bridge the gap between theory and practice? How can the research be used in practice (economic and commercial impact), to influence technical policy, in research (contributing to the body of knowledge)? Add something for field professionals.[Please add]

Limitations of the study:

Please add as heading 6 about the limitations of the study.

Reviewer 2 Report

This is an interesting study that provides a novel insight into the experimental method of investigating simulator sickness. The primary effect that rollover created simulator sickness is to be expected. However, the fact that a number of participants were experiencing symptoms before participating in the main experiment is interesting and was possibly unexpected. Because it was potentially unexpected some of the data that might have been collected to explore the phenomenon does not seem to be available. While this is somewhat a limitation of the current results, I think that without a study like this in the literature there would be limited motivation to study the phenomenon further. Thus, I believe this paper would make a useful contribution to the literature.  

One thing that I found difficult in the abstract was the statement that was used twice - "before the simulation". I think that this needs to be made more clear and explained a bit more in the manuscript as well. "Before the simulation" is vague of how much time before, also presumably there was a health screening so that these people did not have a chronic condition that caused these symptoms. However, what seems essential to me is that at least the time of the SSQ is given so that it isn't as much a puzzle for the reader to solve when looking at the abstract. 

Minor point:

Line 78- It is understandable as it is, but I’d consider age, gender, etc to be a physiological property rather than a physical property.

Reviewer 3 Report

This study is aimed at getting a better understanding of participant-related variables that may influence the symptomatology of simulator sickness using Simulator Sickness Questionnaire. The study involved a 2-minute mobile-platform car rollover simulation conducted in a group of 100 healthy adult participants before and after the simulation. The findings reveal that men suffer from disorientation symptoms more often than women as well as the participants with prior experiences of extra driving training or car accidents.

The paper is clearly written very well and uses a language which is easier to follow. The author has followed well established methodology to conduct the study and answer the well formulated research questions. Results are presented in a very nice way followed by a very good discussion.

This paper is clearly written. Author need to provide slightly more background of the topic that could give the novice reader an appreciation of the use of this technology and successfully link the reader with the topic of study. There exist multiple theories of motion sickness which are not explained well in the context. Also related work is not state of the art and presented briefly. These could be presented in a better way. Also, SSQ is missing that could be provided in the appendix.

I believe this paper is addressing an interesting problem with an effective approach. The presentation is also generally good. Therefore, I would recommend acceptance of this paper, provided issues mentioned above are fixed.

Round 2

Reviewer 1 Report

This manuscript still needs some improvements.

(Comments attached)

This manuscript is a resubmission of an earlier submission. The following is a list of the peer review reports and author responses from that submission.

Round 1

Reviewer 1 Report

This is a well-written article about individual factors related to simulator sickness, which is pretty interesting. Still, I have some questions and suggestions which I will pose below.

Abstract: Please remove the reference from here. Remember that usually the abstract appears separated from the rest of the article.

Introduction: In the last paragraph, authors make reference to previous studies about the influence of participant-related factors and participants' previous experiences in simulator sickness (lines 72-76). However, despite the extensive bibliography they present to the reader, these specific articles are not cited. Why? I think it would be convenient to do that, and justify why the authors' article is different and what is its specific contribution to what is known about this topic. I would also like to see the objective more clearly redacted.

Methods: This section is sound and well-explained.

Results: In Table 1 should also appear the information relative to participants’ prior participation in simulations, in extra driving training or in collisions (which is showed later, lines 166-167).

In Table 3, I feel that it is not necessary to include information relative to median. Also, pay attention to the footnote -SD is not showed in this table, so can be removed from here.

Lines 161-164 makes reference to the differences found between genders, but data are not shown.

In Table 4, I found strange the change of format regarding p value – I suggest that the authors make it uniform with former tables.

Discussion and conclusions: The authors make a good comparison of their results with those in previous studies, and discuss the possible biases in their work. However, I am not sure about the final conclusions. I think they should be rewritten in order to clearly show that they are supported by the results. Now, in the present form, I feel that the obtained results are quite modest to maintain the conclusions about the the possible usefulness of simulators to treat driving-related fears.

Reviewer 2 Report

100 participants rolled over in a motion simulator for vehicles. Before and after their rollover they completed the simulator sickness questionnaire that was developed by Kennedy, Lane, Berbaum, and Lilienthal (1993). The participants scored especially and significantly higher on the questions of the disorientation scale after their rollover than before their rollover. The following between subject factors were distinguished: gender, participants who previously have been in a simulator (of any kind?) (N=11) and participants who have not (N=89), participants who had attended extra driver training (training on top of the regular driving lessons to pass the driving test) (N = 64) and those who had not (N = 36), and participants who had been involved in a car accident or collision (N = 38) and those who had not (N = 62). There were two interaction effects: the ‘pre-test and post test scores on the oculomotor scale’ X  ‘accident involvement’, and the pre-test and post test scores on the disorientation scale’ X ‘accident involvement’. Participants with accident experience scored higher on the oculomotor scale on the pre-test than those who had not been involved in a crash and scored lower on this scale at the post-test than those who had not been involved in a crash.  On the disorientation scale, the participants with prior accident experience scored approximately as high as those who had not on the pre-test but significantly higher on the post test than those who had not.

The paper is well written but the subject is rather trivial. Not surprisingly participants felt more disorientated after they had been turned upside down than before they were turned upside down.

I have quite a few comments:

Abstract. ”Participants with prior experiences of extra driving training or car accidents reported generally more severe SS”. This is not precise. They reported only more distress after the rollover but not before the rollover.

Introduction. In the introduction is mentioned that prior simulator experience, prior accident involvement and extra driver training were between subject factors. They write: Therefore, apart from studying the effect of gender, we also examined the possible influence of the previous participation of study subjects in extra driving training or car accidents.”  However, they do not explain why they selected these between subject factors. With regard to prior simulator experience thy als do not mention what kind of simulator experience this was. More explanation is required. They also asked participants what their age was. Why has age not been included as covariate in the models?

Materials an methods. The authors write: “The study group – 100 participants randomly chosen from the people who volunteered to take part in the simulations offered by the Interactive Safety Center-“. The do not write how these participants were recruited. It is also not mentioned that the test protocol was approved by an ethical committee and that the participants were informed about the test and signed an informed consent form. This is important because it could be expected that some participants would experience motion sickness.

Materials and methods. Line 132. The authors write “Wilcoxon test”. There is more than one Wilcoxon test. They should write “Wilcoxon singed-rank test”.

Materials and methods (and Results). Lines 135-136. The authors write that they consider effect sizes. This is correct. However, they consider only the effect sizes of the mixed ANOVAs (partial eta-squared). However, all the effect sizes of the paired t-tests (Cohen’s d or r) and the effect sizes of the Wilcoxon singed-rank test are missing. Effect sizes need to be included in the tables 2, 3, and 4 of the results section. Regarding these tables. Only the p-values are included but not the test-values and the degrees of freedom.   

Results. Lines 161-163. Only the p-value is provided (p=.003) but not the test something like: ‘t(X)= XX.XX,  p = .003, d = X.XX.

Results. Table 4. The authors first present the results of the differences between subjects in table 4 and then they provide the results of the two significant interaction effects. This is not the correct order. Fisrst they have to provide the results of the mixed ANOVA’s (main effects and interaction effects) and then they have to provide the results of the post hoc tests, the t-tests (both the paired-tests and the independent t-test) when necessary corrected for the problem of multiple comparisons.

Discussion. Lines 202-205. The authors mention as limitation of the study a possible response bias. This is a bias that participants want to please the experimenter and therefore report for instance more disorientation at the post-test than they actually have. However, I think that there also could be another bias. The perspective of being turned around soon in a motion simulator may have caused some stress in advance and this may have influenced the scores on the pre-test.

Reference

Kennedy, R. S., Lane, N. E., Berbaum, K. S., & Lilienthal, M. G. (1993). Simulator Sickness Questionnaire: An Enhanced Method for Quantifying Simulator Sickness. The International Journal of Aviation Psychology, 3(3), 203-220. doi:10.1207/s15327108ijap0303_3

Round 2

Reviewer 2 Report

The article has improved considerably now the effect sizes are included. There is however one thing that concerns me. The authors reply that they cannot specify what extra driving training and what kind of pervious simulator experience the participants have had. They also cannot provide information about the traffic accidents the participants have been involved. This is simply not possible because they did not ask. I consider this as a very serious omission. They write that the aim of their study is to understand what the experiences of participants may have on the development of simulator sickness. This what has not done before and this is what makes their study unique (lines 69-84). However, what has been asked about previous experience was very limited. In my previous review I have forgotten to mention that the study is a before and after study without a control group. This is a weak design. I admit that a control group would have been difficult to realize because it would have meant that participants in the control group should have to complete a questionnaire and after a hour should have to complete the same questionnaire again. However, when you do not do this you will never know what the influence of the completion the first questionnaire has on the experience in the motion simulator and the completion of the second questionnaire.  Therefore my advice to the journal is to reject the article.